# New Circulating Circular RNAs with Diagnostic and Prognostic Potential in Advanced Colorectal Cancer

**DOI:** 10.3390/ijms222413283

**Published:** 2021-12-10

**Authors:** Maria Radanova, Galya Mihaylova, Oskan Tasinov, Desislava P. Ivanova, George St. Stoyanov, Neshe Nazifova-Tasinova, Rostislav Manev, Ayshe Salim, Miglena Nikolova, Diana G. Ivanova, Nikolay Conev, Zhasmina Mihaylova, Ivan Donev

**Affiliations:** 1Department of Biochemistry, Molecular Medicine and Nutrigenomics, Medical University of Varna, 9000 Varna, Bulgaria; galya.mihaylova@mu-varna.bg (G.M.); oskan.tasinov@gmail.com (O.T.); desiplamenova@gmail.com (D.P.I.); neshe.tasinova@mu-varna.bg (N.N.-T.); ayshe.salim@mu-varna.bg (A.S.); miglena.todorova@mu-varna.bg (M.N.); divanova@mu-varna.bg (D.G.I.); 2Clinic of General and Clinical Pathology, University Hospital “St. Marina”, 9000 Varna, Bulgaria; georgi.stoyanov@mu-varna.bg; 3Department of General and Clinical Pathology, Forensic Medicine and Deontology, Division of General and Clinical Pathology, Medical University of Varna, 9000 Varna, Bulgaria; 4Department of Oncology, Medical University of Varna, 9000 Varna, Bulgaria; rostislav.manev@mu-varna.bg (R.M.); nikolay.conev@mu-varna.bg (N.C.); 5Clinic of Medical Oncology, University Hospital “St. Marina”, 9000 Varna, Bulgaria; 6Clinic of Medical Oncology, Military Medical Academy, 1000 Sofia, Bulgaria; zhamina.mihaylova@gmail.com; 7Clinic of Medical Oncology, Hospital “Nadezhda”, 1000 Sofia, Bulgaria; ivan_donev75@abv.bg

**Keywords:** circular RNAs, circSMARcA5, circFUT8, circIPO11, circABCB10, colorectal cancer

## Abstract

Circular RNAs (circRNAs) are a group of special endogenous long non-coding RNAs which are highly stable in the circulation, and, thus, more suitable as new biomarkers of colorectal cancer (CRC). The aim of our study was to explore the plasma expression levels of four circRNAs: has_circ_0001445, hsa_circ_0003028, hsa_circ_0007915 and hsa_circ_0008717 in patients with CRC and to evaluate their associations with clinicopathological characteristics and the clinical outcome of the patients. CircRNAs were extracted from patients’ plasma obtained prior to chemotherapy. Their expression levels were measured by qPCR and calculated applying the 2^−ΔΔCt^ method. The levels of all four circRNAs were significantly increased in the plasma of CRC patients. At the optimal cut-off values hsa_circ_0001445 and hsa_circ_0007915 in plasma could significantly distinguish between patients with or without metastatic CRC with 92.56% sensitivity and 42.86% specificity, and with 86.07% sensitivity and 57.14% specificity, respectively. The mean overall survival (OS) of patients with high/intermediate expression of hsa_circ_0001445 was 30 months, significantly higher in comparison with the mean OS of the patients with low expression—20 months (log-rank test, *p* = 0.034). In multivariate Cox regression analysis, the low levels of hsa_circ_0001445 were also associated with shorter survival (HR = 1.59, 95% CI: 1.02–2.47, *p* = 0.040). A prognostic significance of hsa_circ_0001445 for patients with metastatic CRC was established.

## 1. Introduction

Colorectal cancer (CRC) is one of the most common cancer types in both males and females, worldwide, as well as a leading cause for cancer-related deaths each year [1]. Currently there are no effective non-invasive biomarkers for early diagnosis or established biomarkers for metastatic spread and patient’s outcome prognosis. This, together with the high rate of treatment of refractive CRC, leads to poor clinical outcome in a large number of patients; thus, it is of upmost importance to establish future reliable biomarkers for CRC development, progression, prognosis and treatment. For many years, small non-coding RNAs, mainly miRNAs, were extensively investigated in association with different types of cancer, including CRC [2,3,4]. Their abnormal expression was implicated in CRC development and progression, and the possibility of being detected in different biological samples contributed to current clinical achievements related to miRNA’s role as a diagnostic and prognostic tool [5,6]. The best biomarkers, however, remain to be found. Following miRNA studies, the clinical relevance of long non-coding RNAs (lncRNAs) has also been posited in multiple cancer types regarding their applicability, not only as a new way to understand the fundamental mechanisms of oncogenesis, but mainly as diagnostic, prognostic and predictive tools [7,8,9].

Recently, lncRNAs have been a focus point as non-invasive biomarkers related to CRC, which may increase the number of CRC patients who can benefit from early diagnosis and prognosis [10]. In particular, circRNAs, a type of lncRNAs, have shown promise in this emerging field. They are widely expressed in diverse cell types and can differ depending on the tissue or developmental stage [11]. CircRNAs are highly stable in comparison to linear RNAs due to their covalently closed loop structure, which disables RNase R from binding and cleaving the circRNAs [12]. Multiple studies establish the existence of thousands of circRNAs in peripheral whole blood, as well as in plasma samples alone [13,14]. Some circRNAs are found in larger quantities in cancer cell-produced exosomes and human serum exosomes (exo-circRNAs) and this can be applied to distinguish cancer patients from healthy subjects [15]. Easily accessible sampling for circRNAs measurement confirms their possible applicability as non-invasive biomarkers for CRC.

For the aim of the present study, we selected four circRNAs for investigation: hsa_circ_0001445 (circSMARcA5), hsa_circ_0003028 (circFUT8), hsa_circ_0007915 (circIPO11) and hsa_circ_0008717 (circABCB10). Currently, circIPO11 is being examined in liver cancer [16], circFUT8-in liver and bladder cancer [16,17], circABCB10-in breast, lung and non-small cell lung cancer [18,19,20] and circSMARcA5-in cervical, prostate, hepatocellular, and non-small cell lung cancer, as well as in glioblastoma multiforme [21,22,23,24,25,26,27]. With the exception of circABCB10 none of the above has been previously evaluated in CRC [28,29]. In our study we measured their baseline expression in the circulation of patients with stage III and stage IV CRC before the start of their treatment and in comparison to healthy controls. We investigated the relationship of the circRNA levels with the clinicopathological characteristics and the clinical outcome of the patients. We explored the correlation between hsa_circ_00001445 and hsa_circ_00007915 levels and their respective sponge target microRNAs—miR-181b and miR-106a. These miRNAs, in contrast to the circRNAs that are the focus of our study, have been previously examined in a number of studies in relation to CRC prognosis and diagnostics using plasma or serum samples [30,31,32,33,34,35,36].

## 2. Results

### 2.1. Expressions of hsa_circ_0001445, hsa_circ_0003028, hsa_circ_0007915 and hsa_circ_0008717 Were Increased in Plasma of CRC Patients

The levels of hsa_circ_0001445, hsa_circ_0003028, hsa_circ_0007915 and hsa_circ_0008717 in plasma of CRC patients were significantly increased as compared to healthy controls, with an average-fold change of 1.906, 1.417, 1.734 and 1.789, respectively (Figure 1). The median expression levels of these four circRNAs in healthy controls were: for hsa_circ_0001445, 0.856 (0.568–1.080); for hsa_circ_0003028, 0.830 (0.758–0.927); for hsa_circ_0007915, 0.672 (0.595–0.744); and for hsa_circ_0008717, 0.922 (0.845–1.024). To evaluate the sensitivity and specificity of each circRNA as a potential diagnostic marker, ROC curve analyses were performed (Figure 1). The AUC values and optimal cut-off in the largest Youden index for each circRNA are summarized in Table 1. All circRNAs have a statistically significant accuracy as independent diagnostic markers, with the most sensitive one being hsa_circ_0007915, with AUC = 0.776 (95% CI: 0.71–0.84, *p* < 0.0001) (Figure 1C, Table 1).

### 2.2. Comparison between circRNA Plasma Levels and the Clinicopathological Characteristics of the CRC Patients 

The Mann–Whitney U test was performed to compare the plasma levels of the investigated circRNAs in patients with different clinicopathological characteristics. The expression levels of hsa_circ_0001445 and hsa_circ_0007915 in plasma were significantly different in stage III vs. stage IV CRC patients (*p* = 0.0001, *p* = 0.0003, respectively, Table 2). A high level of hsa_circ_0003028 expression was found in patients with a primary tumor location in the left part of the colon vs. the right one (*p* = 0.021, Table 2). There was also a trend between higher levels of hsa_circ_0003028 expression and a low histological grade and a presence of *RAS* mutations, and between higher levels of hsa_circ_0007915 and CEA > 2 ULN (Table 2). There were no significant differences between the expressions of the investigated circRNAs and the other clinicopathological parameters (patients’ ages, sex, PS (ECOG), presence of liver and other tissues metastasis, and the appearance of a local recidive) (Table 2). 

### 2.3. Hsa_circ_0001445 and hsa_circ_0007915 can Distinguish CRC Patients in Stage III from Patients in Stage IV 

When CRC patients in stage III were compared with patients in stage IV, we found that only hsa_circ_0004515 and hsa_circ_0007915 could significantly distinguish between the CRC stages (Figure 2). The AUC for these two circRNAs were 0.729 (95% CI: 0.62–0.84, *p* = 0.0002) and 0.714 (95% CI: 0.59–0.84, *p* = 0.0004), respectively (Figure 2). At the optimal cut-off value for hsa_circ_0004515 based on the largest Youden index 0.354, the sensitivity was 92.56% and the specificity was 42.86% (Figure 2). The largest Youden index for hsa_circ_0007915 was 0.432 and at the optimal cut-off value the sensitivity and specificity were 86.07% and 57.14%, respectively (Figure 2).

To compare the accuracy of hsa_circ_0001445 and hsa_circ_0007915 as diagnostic tests we evaluated the difference between the AUCs of both ROC curves. The analysis did not indicate a significant difference between these two potential markers, and the tests were equivalent (Difference = 0.015, St. error = 0.837, Z-statistic = 0.179, *p* = 0.858). The correlation analysis showed a weak positive correlation (r = 0.266, *p* = 0.001) between the plasma levels of hsa_circ_0004515 and hsa_circ_0007915 in CRC patients.

### 2.4. Hsa_circ_0001445 and hsa_circ_0007915 have Higher Expression in TME

To compare differential gene expression of hsa_circ_0001445 and hsa_circ_0007915 in the tumor and in the tumor microenvironment (TME) we used Tumor Immune Single Cell Hub (TISCH) [37]. Among the available human CRC datasets (i.e., CRC_GSE108989, CRC_GSE136394, CRC_GSE139555, CRC_GSE146771_10X, CRC_GSE146771_Smartseq2) we chose CRC_GSE146771_Smartseq2, because, in contrast to the others, it contained single-cell transcriptome profiles not only from immune and stromal cells but also from malignant ones [38]. Figure 3 shows that expression levels of both circRNAs in malignant cells are lower than in stromal cells; hsa_circ_0001445 expression is higher in stromal cells than in immune cells and hsa_circ_0007915 expression is highest in stromal cells. The expression of both circRNAs is higher in the TME than in the tumor cells. More detailed analyses of the transcriptome profiles of these circRNAs in TISCH established that expression of hsa_circ_0001445 in immune cells was highest in proliferative T cells, followed by that in T regs, CD8+ T cells and in NK cells. Amongst stromal cells, its expression in fibroblast and endothelial cells was highest. The expression of hsa_circ_0007915 was highest only in endothelial cells.

### 2.5. Low hsa_circ_0001445 Levels in Plasma Are Associated with Short Survival

We have found that of all investigated circRNAs only hsa_circ_0001445 had a prognostic significance (Figure 4). Patients were divided into three groups according to their expression levels of hsa_circ_0001445-with low (up to the 33rd percentile, range: 0.13–1.20), with intermediate (between the 33rd and 66th percentiles, range: 1.21–2.21), and with high expression (over the 66th percentile, range: 2.22–6.86). In the group of CRC patients in stage IV, those with low expression of hsa_circ_0001445 had a mean overall survival of 20 months (95% CI: 16.70–23.76), those with high expression of hsa_circ_0001445 had a mean overall survival of 26 months (95% CI: 20.23–32.59), and the patients with intermediate expression of hsa_circ_0001445 had a mean overall survival of 33 months (95% CI: 26.02–40.54). In a subsequent analysis we combined the two groups with high and intermediate expression of hsa_circ_0001445 based on their similar higher mean overall survival compared to the patients with low expression (Figure 4). The mean overall survival of this group was established to be 30 months (95% CI: 24.99–34.73), significantly higher in comparison with the mean overall survival of the patients with low expression of hsa_circ_0001445—20 months (log-rank test, *p* = 0.034, Figure 4).

After each variable was separately entered in a Cox’s model, the variables that had a *p* ≤ 0.1 were entered into a multiple Cox’s regression model (Table 3). In univariate and also in multivariate Cox regression analysis the low levels of hsa_circ_0001445 were also associated with a shorter survival of the patients (HR = 1.58, 95% CI: 1.02–2.45, *p* = 0.039; HR = 1.59, 95% CI: 1.02–2.47, *p* = 0.040, Table 3). By Cox regression analysis, left primary tumor location (HR = 0.57, 95% CI: 0.36–0.91, *p* = 0.019; HR = 0.61, 95% CI: 0.38–0.97, *p* = 0.036, Table 3) and CEA ≤ 2 ULN (HR = 0.51, 95% CI: 0.32–0.80, *p* = 0.003; HR = 0.52, 95% CI: 0.33–0.80, *p* = 0.004, Table 3) were defined as independent predictors for longer overall survival of our metastatic CRC patients.

### 2.6. Expressions of miR-181b as a Target for hsa_circ_0001445 and of miR-106a as a Target for hsa_circ_0007915 Are Increased in the Plasma of the CRC Patients

Many studies observed the functional mechanisms of circRNAs to modulate the pathological processes of cancer and the main one among these was via sponging of an miRNA or miRNAs. According to Ren et al. (2017) and Yu et al. (2018), miR-106a and miR-181b are sponged by hsa_circ_0007915 and hsa_circ_0001445, respectively [9,17]. If these miRNAs are targets for circRNAs, their levels would be dysregulated. To test this hypothesis we examined the expression of miR-181b and miR-106a in our CRC patients. These analyses were performed only for stage IV patients (*n* = 62) and for HC. We found that both miRNAs were differentially expressed at significantly higher levels in the plasma of CRC patients and not in healthy controls (*p* < 0.0001, Figure 5a; *p* < 0.0001, Figure 5b). Analyses of the ROC curve confirmed that the expression of the two miRNAs in plasma distinguished CRC patients in stage IV from healthy controls (Figure 5c,d). Comparing the two miRNAs we found that miR-181b exhibited greater diagnostic power with an AUC value of 0.891 (95% CI: 0.84–0.94, *p* < 0.0001, Figure 5c), with 88.24% sensitivity, and with 83.33% specificity at Youden Index 0.716 than the miR-106a with an AUC = 0.774 (95% CI: 0.70–0.85, *p* < 0.0001, Figure 5d), with 84.38% sensitivity and 64.44% specificity at larger Youden Index 0.488.

We performed a correlation analysis to see whether the levels of miRNAs in plasma would correlate to the levels of circRNAs. No significant correlations were found between the plasma level of miRNAs and circRNAs (Figure 6a,b). As shown in Figure 6a, there was a trend of weak negative correlation between the levels of miR-181b and hsa_circ_0001445. This result provides a possibility, albeit indirectly, of hsa_circ_0001445 sharing binding sites with miR-181b.

## 3. Discussion

Finding noninvasive diagnostic, prognostic or predictive biomarkers in the blood is a constant target in cancer studies. In recent years great hopes have been placed on lncRNAs and, especially, on circRNAs because of their association with clinicopathological characteristics of cancer patients and their stability in circulation. Similar to the other types of ncRNAs, circRNAs can act as oncogenes or tumor suppressor genes and are associated with cancer phenotypes [29,39].

In our study we found that expression of the four circRNAs—hsa_circ_0001445, hsa_circ_0003028, hsa_circ_0007915 and hsa_circ_0008717—was significantly increased in the plasma of patients with advanced disease as compared to healthy controls. Furthermore, we found that two of those, hsa_circ_0004515 and hsa_circ_0007915, could significantly distinguish between patients in stage III and patients in stage IV with a moderate accuracy with AUC = 0.729 and AUC = 0.714, respectively. With regard to hsa_circ_0001445, our results allow us to identify it as a novel prognostic marker for metastatic CRC patients. We found that a panel of hsa_circ_0001445 and hsa_circ_0007915 had a robust diagnostic value as a means to differentiate between patients in stage III from those in stage IV (90.98% sensitivity and 60.71% specificity).

There is only one study on hsa_circ_0007915 in liver cancer [16], which does not allow us to compare nor comment on our results in parallel, since the authors have used a small number of samples and have not provided important clinical information about their patients. Studies on hsa_circ_0001445 in solid tumors show its essential role in the development and progression of cancer. CircSMARCA5 is being investigated mainly in tumor tissues [21,22,23,25,26,27]. There are only three studies that, along with the expression of hsa_circ_0001445 in the tumor tissue, have also measured its expression in plasma [24,40,41]. Hsa_circ_0001445 was established as a circRNA with tumor suppressive effects in hepatocellular carcinoma [24,25,41], gastric cancer [40], non-small cell lung cancer [26] and glioblastoma multiforme [27]. For instance, in prostate cancer, hsa_circ_0001445 was shown to act as an oncogene [23]. Moreover, regarding cervical cancer, there are two contradictory studies—one attributing to hsa_circ_0001445 the role of an inhibitor of proliferation, migration and invasion and an inductor of the cell cycle arrest [21], and the other study claiming that hsa_circ_0001445 plays an important role in the progression of cervical cancer via the ERK signaling pathway [22]. Similar to our findings, there are reports about the prognostic value and an established correlation of low expression levels of circSMARCA5 with a poorer overall survival reported in gastric cancer [40] and in hepatocellular carcinoma [25].

Since most studies on circRNAs review the tumor tissue expression of circRNAs and do not represent their plasma levels, there is an unmet need for studies to examine the levels of circulating circRNAs. Interestingly, studying these poorly explored ncRNA expressions in plasma we found a tendency for most of these circulating ncRNAs to be highly expressed in patients with a metastatic CRC, and, at the same time, these levels were related to a good prognosis. This fact raises several questions regarding circulating ncRNAs, such as: which is their source; is their expression related to the disease stage; are they reliable biomarkers?

Multiple studies pay attention to circRNA biogenesis inside the cell, but overall literature data about non-coding RNA sources in cancer are scarce and this limitation makes it difficult to explain when ncRNA expression differs in the tissue and in the plasma. The main hypothesis is that circulating ncRNAs are products of the tumor and the TME. They are packed in particles such as microvesicles, exosomes, prostasomes and apoptotic bodies, and then are either actively secreted into the circulation and probably taken up by the recipient cells [42] or they are passively released during tissue injury, chronic inflammation, apoptosis and necrosis [43,44]. Few studies examine the cells of TME as a source of ncRNAs. Kristensen et al. (2020) demonstrate that high levels of ciRS-7 (hsa_circ_0001946, hsa_circRNA_105055, CDR1as) in CRC samples are not due to overexpression in malignant cells, as previously thought, but rather to an abundance in the stromal cells within the tumor [45]. Gu et al. (2020) report about circSLC7A6 (hsa_circ_0039943) exosome secretion in CRC from cancer-associated fibroblasts (CAFs), the main component of the stroma of tumor [46]. These stroma cells support tumor progression (metabolism, proliferation, metastasis, anti-apoptosis, angiogenesis and therapy resistance) via direct/indirect collaboration with the nearby malignant cells [47]. Zhou et al. (2021) also comment on CAFs as lncRNAs sources in CRC [48]. Moreover, according to Luo et al. (2020) exosomal lncRNAs are delivered from the immune cells to the malignant cells and to other immune cells in the TME [49]. If this is the case, it is of a great importance to define the source of the ncRNAs, since their role in the immune response during tumorigenesis depends on whether they are derived from tumors or from immune cells [50]. Furthermore, the area of circRNA expression in the immune cells involved in the TME is less clear at present.

In our current study we have used the TISCH database, including single-cell transcriptome profiles, to present the detailed differential gene expression of hsa_circ_0001445 (SMARCA5) and hsa_circ_0007915 (IPO11) in malignant cells and in immune and stromal cells of the TME. Lower expression was established in the tumor cells in comparison with that in the cells of the TME. If the source of hsa_circ_0001445 were stromal cells and immune cells and of hsa_circ_0007915 mainly stromal endothelial cells, this could explain their high levels in plasma. This could also explain the differences in the expression levels in the peripheral plasma of the two target groups of patients given that stage IV tumors—those with a larger primary size, more lymph node metastasis, and multiple and larger distant metastases—would have a proportionally larger overall tumor size, hence more of these cells from the overall larger TME would spontaneously and through necrosis release the circRNAs into the circulation [51].

Our hypothesis is that these circRNAs probably have tumor suppressor effects and their expression is inhibited in malignant cells; therefore, the cells of the TME are expressing them strongly (Figure 7).

The levels of hsa_circ_0001445 and hsa_circ_0007915 and of the other investigated circRNAs were highest in metastatic CRC patients due to proportionally larger overall tumor size and its interaction not only with the TME-immune response, neovascularization, tumor cell necrosis and vascular induced necrosis of large parts of the primary tumor, but also with the resident immune cells in the organs with metastasis (i.e., Kupffer cells in the liver) [52,53,54,55]. As such, not only would the levels of these circRNA be higher in the peripheral plasma of patients with malignancy whilst compared with healthy controls, but they would also show differentiation—higher in patients with stage IV malignancy as compared to those in stage III and higher in patients with poorly differentiated (G3) versus intermediately differentiated (G2 and G1) malignancy due to the extent of the TME reaction and an enhanced necrosis [52,56,57].

Plasma and tissue expressions of ncRNAs in solid tumors are not infrenquently different. Little is known about the reasons for this and whether perpetually circulating circRNAs reflect the conditions in the tumor, especially if they originate from a different source. Upregulated ncRNAs usually have oncogenic functions as factors promoting cell proliferation, invasion/migration, metastasis and cell cycle progression while at the same time decreasing apoptosis rate [58]. If this can always be applied to the upregulated ncRNAs in the circulation is yet to be determined. If we consider circRNAs as a special group of ncRNAs, we must pay attention to another phenomenon and try to find its place in the whole puzzle. Sun et al. (2021) suggest a new model for circRNA structure based on the existence of internal complementary base-pairing sequences in plenty of circRNAs [59]. The hypothesis is described in the so called “open–close effect” that may play a greater role than solely the change in expression levels. Thus, if higher expressed circRNAs are in a closed state, they do not exert biological effects; and the opposite—if normal/lower expressed circRNAs are in an open state, they can exert their biological functions. Moreover, if a given circRNA has binding sites for both tumor suppressor and oncogenic miRNAs, with reference to this model it might selectively “open” or “close” these binding sites and so give rise to different effects.

The model of Sun et al. (2021) may explain why there are no interactions with miRNAs, which were in silico predicted as circRNA targets [59]. According to the miRNet database [60], hsa_circ_0001445 interacts in peripheral blood with hsa-miR-19a-3p, hsa-miR-19b-3p and hsa-miR-33a-5p, while hsa_circ_0007915 interacts with hsa-miR-155-5p and hsa-miR-223-3p. However, we tested a part of our samples for correlations between already studied interactions between hsa_circ_0001445 and miRNA-181b and between hsa_circ_0007915 and miRNA-106a. We found a tendency for interactions only with regard to the first couple. Studies with larger sample cohorts and studies on other circRNA-miRNA interactions are needed to confirm or reject the established trend.

Our study has certain limitations that have to be acknowledged. The study is descriptive and raises questions about the functions and origins of explored circRNAs in CRC. We made a speculative analysis about the source of hsa_circ_0001445 and hsa_circ_0007915 and presented a hypothesis—using the TISCH database—of the possible reasons for elevated levels of circRNAs in advanced CRC disease. Despite the listed limitations, the study provides new data on plasma expression levels of circRNAs in a relatively large homogeneous group of 122 patients with metastatic CRC, all investigated prior to treatment.

## 4. Materials and Methods

### 4.1. Patients and Healthy Controls

The patients were selected and followed up at the Military Medical Academy, Sofia and at the University Hospital “St. Marina“, Varna. The study included 122 patients with stage IV and 28 patients with stage III CRC per the American Joint Committee on Cancer (AJCC) Cancer Staging Manual, 8th ed. [35]. Patients with stage IV were unresectable and their therapies included first-line fluoropyrimidine-based CT (FCT) or FCT in combination with anti-VEGF or with anti-EGFR targeted therapy. Patients with stage III CRC had undergone radical surgery, were without residual disease and completed 5-FU based adjuvant chemotherapy. Peripheral blood was obtained from all selected patients before starting the treatment.

The stage IV patients’ group consisted of 80 males (65.6%) and 42 females (34.4%) with a mean age at the time of diagnosis 60.3 ± 11.51. At the time of diagnosis 89 patients (73%) had liver metastasis combined with at least one more extra-hepatic metastatic site. Extra-hepatic only metastases were detected in 33 patients (27%). The lungs were the most common extra-hepatic metastatic site (26 patients, 78.7%) followed by peritoneal metastases (5 patients, 15.1%) and distant lymph node metastases (2 patients, 6.2%). With respect to primary tumor location, 94 (77%) patients were diagnosed with tumor of the left colon and 28 (23%) patients of the right colon. The subgroup analysis also included patients’ *RAS* mutation status. A total of 57 (46.7%) of the patients were positive for *RAS* mutations and 59 (48.4%) had wild type *RAS* mutation status. For 6 (4.9%) patients there was no information for a presence of *RAS* mutations. Tumor differentiation was also included in the subgroup analysis. Well-differentiated G1 tumors totaled 9 (7.4%), moderately differentiated G2 tumors totaled 89 (73.0%) and poorly differentiated G3 totaled 24 (19.6%). The patients’ Eastern Cooperative Oncology Group (ECOG) performance status was assessed to be <2. The stage III patients’ group consisted of 15 males (53.6%) and 13 females (46.4%) with a mean age at the time of diagnosis 63 ± 8.89. Subjects with left colon cancer were 20 (71.4%); 8 (28.6%) were diagnosed with primary tumor of the right colon. With respect to tumor differentiation, 25 (89.3%) patients had moderately differentiated G2 tumors and 3 (10.7%) patients had well-differentiated G1 type tumors. The patients’ ECOG performance status observed during the adjuvant chemotherapy was ≤1. Serum levels of carcinoembryonic antigen (CEA) were assessed before the first cycle of chemotherapy in all included patients. A total of 90 healthy volunteers, age and sex matched to the patients, were included as a control group for analyses in the study. The characteristics of participants are summarized in Table A1.

This study has two approvals of the “Commission for Scientific Research Ethics” of the Medical University of Varna, Bulgaria (protocol №74/03.05.2018 and protocol №83/16.05.2019). It was conducted on leftover plasma samples from patients with metastatic CRC and on prospectively collected plasma from CRC patients in stages III and IV. One section of the patients (*n* = 101) signed an informed consent for scientific research for the initial bio-bank and data collection according to the Declaration of Helsinki (protocol №74/03.05.2018) while the other section (*n* = 49)—as well as each selected healthy control volunteer—signed an informed consent form prior to enrollment in the current study (protocol №83/16.05.2019). The authors declared that the research was conducted in accordance with the regulations which were approved by the ethical standards of the “Commission for Scientific Research Ethics” of the Medical University of Varna, Bulgaria, and with the principles of the World Medical Association Declaration of Helsinki “Ethical Principles for Medical Research Involving Human Subjects”.

### 4.2. Serum miRNA Extraction

A small RNA fraction was extracted from 300 µL serum samples and treated with 20 µL Proteinase K for 37 °C for 10 min using NucleoSpin miRNA Plasma (Macherey-Nagel, Düren, Germany). RNA extraction continued according to the manufacturer instructions with the following minor modifications: to adjust binding conditions, 500 µL isopropanol was added to the processed sample; to increase miRNA concentration, the column was incubated for 5 min and finally the RNA was eluted with 22 µL RNAse-free water.

### 4.3. Serum Total RNA Extraction and RNAse R Treatment

For the analyses of circRNA levels, total RNA was extracted from 200 µL of serum samples using Monarch Total RNA Miniprep Kit (New England, Ipswich, MS, USA). Extraction proceeded according to the manufacturer instructions. Minor modifications were applied including addition of a higher volume (500 µL) of isopropanol to each sample prior to loading to the purification column. DNAse I treatment was done according to the manufacturer recommendations. RNA elution with 40 µL of nuclease free water was preceded by additional centrifugation at 16,000 g/min RNAse R (10 U/µL) (ABM, Richmond, BC, Canada); treatment was performed to 10 µL of each sample in order to remove the linear RNA molecules prior to subsequent analyses of circRNAs.

### 4.4. cDNA Synthesis and Pre-Amplification

cDNA synthesis was done according to the RevertAid First Strand cDNA Synthesis Kit protocol (Thermo Scientific, Waltham, MS, USA) using 90 ng of RNA template for both miRNAs and circRNAs. Specific stem loop primers and an U6 reverse primer mix and a mix of oligo dT and random hexamer primers were used for the cDNA synthesis of miRNAs and of circRNAs, respectively. Preamplification of 5 µL (miRNAs) or of 3 µL (circRNAs) cDNA template was performed using 2× RedTaq polymerase (Genaxxon Bioscience, Ulm, Germany) and a miRNA For/Universal Rev primer (10 µM) mix or a circRNA For/Rev primer (10 µM) mix. The sequences of all used primers are shown in Table 4.

### 4.5. qPCR

qPCR was performed with ABI 7500 Real Time PCR System (Applied Biosystems, Waltham, MS, USA) using AmplifyMe SG Universal Mix (AM02, BLIRT, Gdańsk, Poland) with 0.5 µL of pre-amplified cDNA template at the following conditions: 50 °C/2 min, 95 °C/5 min; 43 cycles on 95 °C/15 s, 60 °C/45 s; melting curve. As endogenous controls, U6 and β-actin were used for miRNAs and circRNAs, respectively. Used For/Rev primer sequences for miRNAs and circRNAs are presented in (Table 4). All qPCR analyses were performed in triplicates. The relative expression levels were calculated by the 2^−ΔΔCt^ method [61].

The TISCH [37], an online database integrating single-cell transcriptomic profiles and focusing on the TME, was used to present the expression of circRNAs in tumor cells and in TME cells from human CRC patients.

### 4.6. Statistical Analysis

Data were managed and analyzed using GraphPad Prism 6 software. The Mann–Whitney U test was used to compare expression levels of investigated circRNAs in plasma of healthy controls and CRC patients and to evaluate the association of circRNAs with clinicopathological characteristics of patients. Diagnostic accuracy and specificity and sensitivity of each circRNA to distinguish CRC patients from healthy controls and CRC in different stages were estimated by using Receiver Operating Characteristics (ROC) analysis and Youden indices. Tests with area under the curve (AUC) values greater than 0.9 had high accuracy, those between 0.9–0.7 had moderate accuracy and those between 0.6–0.5 had low accuracy [62]. Hazard ratios (HRs) and 95% confidence intervals (CIs) for univariate and multivariate models were calculated using Cox proportional-hazards regression models. Survival curves were estimated using the method of Kaplan–Meier and differences were assessed using the log-rank test. Spearman correlation was used to evaluate the correlations between circRNAs and their potential miRNAs targets. Two-tailed *p*-values (<0.05) were considered significant.

## 5. Conclusions

In conclusion, this is the first study evaluating plasma levels of hsa_circ_0001445, hsa_circ_0003028 and hsa_circ_0007915 in CRC patients. We found that all investigated circRNAs had certain diagnostic value as independent markers in plasma, but hsa_circ_0007915 had the best diagnostic performance. Hsa_circ_0001445 and hsa_circ_0007915 showed significant expression differences between patients in stage III and stage IV. However, only hsa_circ_0001445 had prognostic significance for metastatic patients. The effects of treatment on the expression levels of circRNAs would be interesting to explore to further reveal the origin, role and mechanism of action of these ncRNAs in CRC.

## Figures and Tables

**Figure 1 ijms-22-13283-f001:**
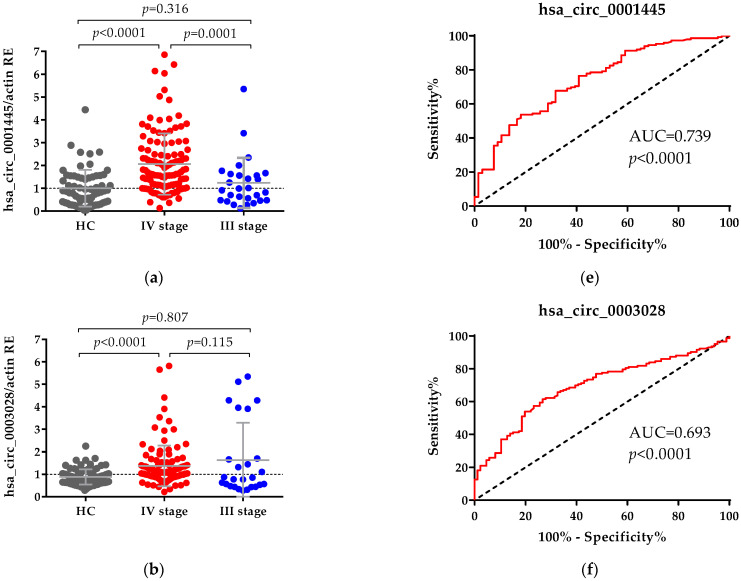
Relative expression (RE) of circRNAs: (**a**) comparison between levels of hsa_circ_0001445, (**b**) hsa_circ_0003028, (**c**) hsa_circ_0007915 and (**d**) hsa_circ_0008717 in plasma samples from healthy controls (HC) and patients with CRC in stages III and IV. The expression levels of all circRNAs were measured using qPCR. β-actin RNA was used as an internal control. Relative gene expression level was calculated using the 2^−ΔΔCt^ method. The Mann–Whitney U test was used to compare HC and CRC patients; data are presented as mean ± SD. ROC curves of (**e**) hsa_circ_0001445, (**f**) hsa_circ_0003028, (**g**) hsa_circ_0007915 and (**h**) hsa_circ_0008717 differentiate patients with CRC from HC.

**Figure 2 ijms-22-13283-f002:**
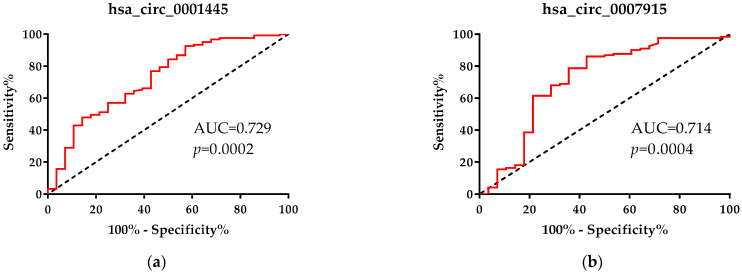
ROC analysis and ROC curves using (**a**) hsa_circ_0001445 and (**b**) hsa_circ_0007915 to differentiate patients with CRC in stage IV from patients in stage III.

**Figure 3 ijms-22-13283-f003:**
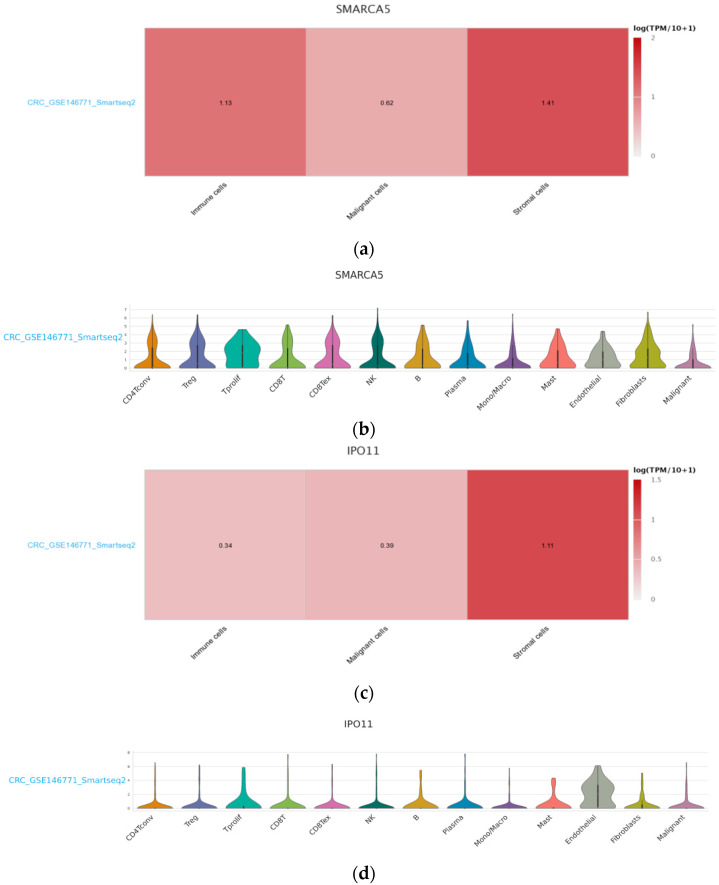
(**a**) Differential gene expression of hsa_circ_0001445 (*SMARCA5*) in malignant cells and immune and stromal sells; (**b**) Detailed presentation of expression levels of *SMARCA5* gene in TME; (**c**) Differential gene expression of hsa_circ_0007915 (*IPO11*) in malignant cells and immune and stromal cells; (**d**) Detailed presentation of expression levels of *IPO11* gene in TME. Visualizations were generated using TISCH database including single-cell transcriptome profiles [37].

**Figure 4 ijms-22-13283-f004:**
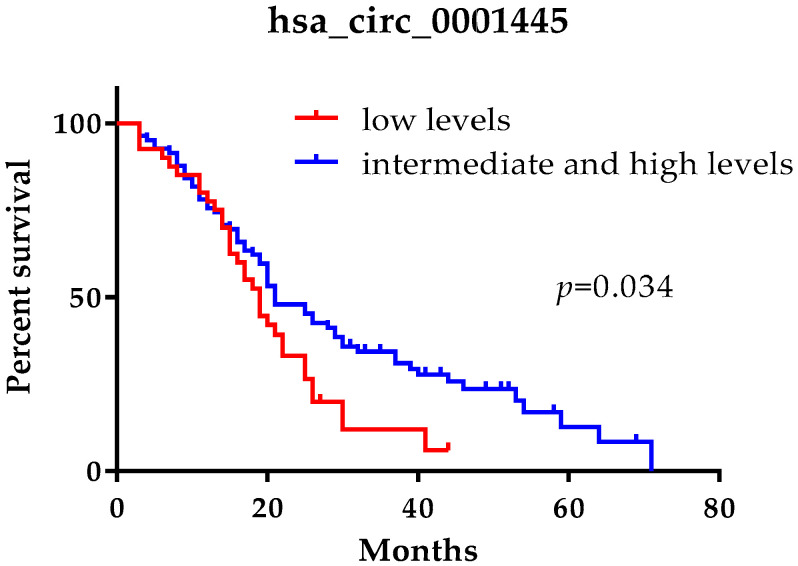
Kaplan–Meier survival analysis for assessment of hsa_circ_0001445 levels and the overall survival of CRC patients in stage IV. Overall survival of patients with intermediate and high levels of hsa_circ_0001445 was compared to overall survival of patients with low levels of the circRNAs.

**Figure 5 ijms-22-13283-f005:**
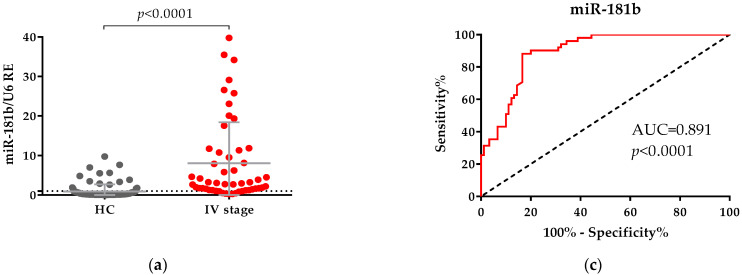
Comparison between levels of (**a**) miR-181b and (**b**) miR-106a in plasma samples from healthy controls (HC) and patients with CRC in stage IV. The expression levels of both miRNAs were measured using qPCR in plasma from CRC patients and HC. U6 RNA was used as an internal control. Relative gene expression (RE) was calculated using the 2^−ΔΔCt^ method. The Mann–Whitney U test was used to compare the two groups—HC and CRC patients; data are presented as mean ± SD. ROC curves of (**c**) miR-181b and (**d**) miR-106a used to differentiate patients with metastatic CRC from HC.

**Figure 6 ijms-22-13283-f006:**
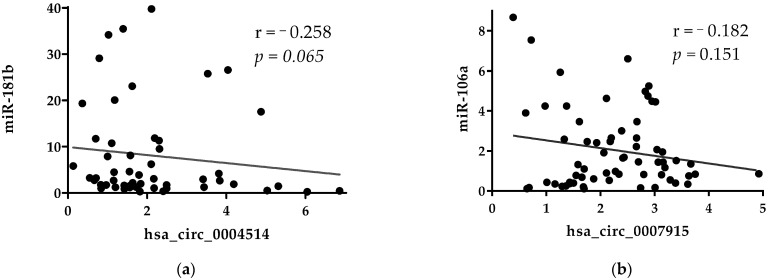
(**a**) Correlation between the expression levels of miR-181b and hsa_circ_0001445 in plasma of CRC in stage IV. (**b**) Correlation between expression levels of miR106a and hsa_circ_0007915 in plasma of CRC in stage IV. Spearman correlation was used to identify the correlation between the levels of miRNAs and circRNAs in plasma.

**Figure 7 ijms-22-13283-f007:**
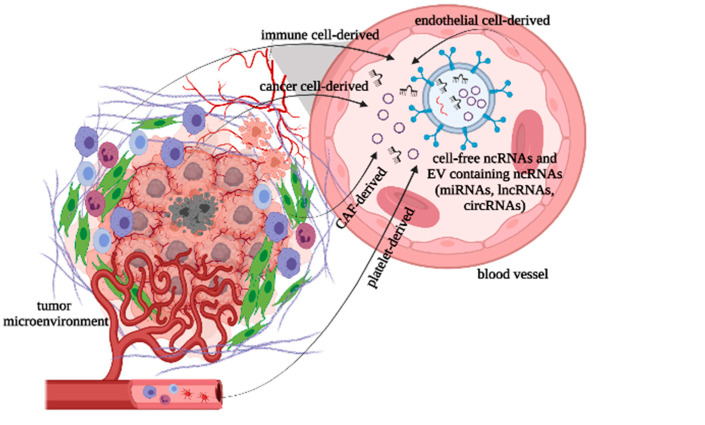
Possible sources of circRNAs in malignancy, based on the limited number of existing studies. Created in biorender.com.

**Table 1 ijms-22-13283-t001:** Characteristics of circRNAs as a diagnostic biomarker.

CircRNAs	Cut-off	AUC (95% CI)	Youden Index	Sensitivity%	Specificity%	*p*-Value
hsa_circ_0001445	1.117	0.739 (0.67–0.81)	0.360	67.79	68.18	<0.0001
hsa_circ_0003028	1.022	0.693 (0.62–0.76)	0.348	61.54	73.26	<0.0001
hsa_circ_0007915	1.239	0.776 (0.71–0.84)	0.464	61.33	85.06	<0.0001
hsa_circ_0008717	1.450	0.626 (0.56–0.70)	0.312	41.22	90.00	0.002

**Table 2 ijms-22-13283-t002:** circRNAs plasma expression and clinicopathological characteristics of the CRC patients.

Clinicopathological Characteristics (Number)	hsa_circ_0001445 Median (95% CI)	*p*-Value	hsa_circ_0003028 Median (95% CI)	*p*-Value	hsa_circ_0007915 Median (95% CI)	*p*-Value	hsa_circ_0008717 Median (95% CI)	*p*-Value
Age <65 (87) ≥65 (63)	1.63 (1.41–2.03) 1.42 (1.15–1.67)	0.136	1.07 (0.98–1.23) 1.14 (1.02–1.35)	0.432	1.43 (1.20–1.87) 1.57 (1.28–2.00)	0.977	1.23 (1.04–1.56) 1.04 (0.86–1.48)	0.694
Sex Female (55) Male (95)	1.55 (1.12–2.11) 1.58 (1.41–1.81)	0.926	1.15 (0.98–1.41) 1.07 (1.02–1.23)	0.343	1.50 (1.16–2.00) 1.53 (1.22–1.87)	0.910	1.31 (0.93–1.70) 1.21 (0.92–1.36)	0.215
Primary tumor location left colon (114) right colon (36)	1.57 (1.41–1.72) 1.66 (0.79–2.30)	0.708	1.15 (1.06–1.25) 0.88 (0.71–1.12)	0.021	1.63 (1.33–1.96) 1.41 (0.98–1.87)	0.295	1.22 (1.04–1.48) 1.02 (0.80–1.81)	0.960
Histological grade G1-G2 (123) G3 (27)	1.57 (1.41–1.79) 1.56 (1.11–2.18)	0.871	1.13 (1.06–1.25) 0.98 (0.85–1.16)	0.054	1.65 (1.36–1.89) 1.23 (0.90–1.96)	0.329	1.23 (1.04–1.54) 0.94 (0.80–1.50)	0.415
PS (ECOG) 0 (59) 1 (91)	1.62 (1.11–2.00) 1.56 (1.39–1.81)	0.786	1.09 (0.91–1.32) 1.12 (1.03–1.25)	0.574	1.37 (1.01–2.20) 1.53 (1.36–1.83)	0.824	1.22 (0.89–1.54) 1.21 (0.93–1.55)	0.965
CEA ≤2 ULN (71) >2 ULN (73)	1.42 (1.15–1.76) 1.63 (1.41–2.17)	0.211	1.14 (0.98–1.35) 1.09 (1.02–1.24)	0.627	1.37 (0.94–1.89) 1.72 (1.43–2.11)	0.087	1.13 (0.89–1.40) 1.37 (1.05–1.70)	0.516
TNM stage IV (122) III (28)	1.65 (1.51–2.10) 0.95 (0.56–1.56)	0.0001	1.12 (1.06–1.24) 0.81 (0.53–1.66)	0.115	1.71 (1.45–1.96) 0.75 (0.62–1.24)	0.0003	1.22 (0.95–1.48) 1.22 (0.75–1.90)	0.942
RAS M+ (57) WT (59)	1.70 (1.42–2.24) 1.58 (1.16–2.10)	0.439	1.25 (1.11–1.35) 1.05 (0.98–1.16)	0.082	1.88 (1.55–2.20) 1.66 (1.28–2.00)	0.410	1.62 (1.09–1.83) 0.99 (0.87–1.33)	0.114
Local recidive yes (20) no (102)	1.79 (1.39–2.49) 1.64 (1.42–2.10)	0.743	1.09 (0.94–1.23) 1.14 (1.06–1.25)	0.565	1.96 (1.28–2.70) 1.69 (1.43–1.92)	0.260	1.37 (0.83–1.83) 1.19 (0.93–1.40)	0.726
Liver metastasis yes (101) no (20)	1.64 (1.41–2.03) 2.14 (1.48–3.07)	0.230	1.14 (1.03–1.25) 1.12 (0.98–1.89)	0.613	1.76 (1.53–2.11) 1.37 (0.90–1.96)	0.185	1.24 (0.93–1.51) 1.09 (0.77–2.04)	0.858
Other metastasis yes (58) no (63)	1.63 (1.48–2.10) 1.87 (1.41–2.30)	0.428	1.10 (1.02–1.23) 1.16 (1.03–1.35)	0.356	1.85 (1.43–2.20) 1.61 (1.28–1.96)	0.227	1.13 (0.84–1.62) 1.28 (0.91–1.58)	0.620

M+—positive for *RAS* mutations; WT—wild type; PS (ECOG)—performance status (Eastern Cooperative Oncology Group); ULN—upper limit of normal; CEA—Carcinoembryonic antigen; the Mann–Whitney *U* test was applied.

**Table 3 ijms-22-13283-t003:** Results of Cox regression analysis for predicting overall survival.

Variable	Univariate Analysis			Multivariate Analysis		
	Hazard Ratio	95% CI	*p*-Value	Hazard Ratio	95% CI	*p*-Value
Age <65 vs. ≥65	1.12	0.73–1.73	0.59	-	-	-
Sex male vs. female	0.895	0.58–1.37	0.61	-	-	-
Histological grade low vs. high	0.72	0.43–1.19	0.20	-	-	-
RAS status WT vs. M+	0.71	0.46–1.11	0.12	-	-	-
Liver metastasis no vs. yes	0.78	0.43–1.37	0.38	-	-	-
Peritoneum metastasis no vs. yes	0.70	0.40–1.23	0.22	-	-	-
Lung metastasis no vs. yes	0.95	0.59–1.52	0.83	-	-	-
Primary tumor location left colon vs. right colon	0.57	0.36–0.91	0.019	0.61	0.38–0.97	0.036
CEA ≤2 ULN vs. >2 ULN	0.51	0.32–0.80	0.003	0.52	0.33–0.80	0.004
hsa_circ_0001445 low vs. high/intermediate	1.58	1.02–2.45	0.039	1.59	1.02–2.47	0.040

M+—positive for *RAS* mutations; WT—wild type; PS (ECOG)—performance status (Eastern Cooperative Oncology Group); ULN—upper limit of normal; CEA—Carcinoembryonic antigen.

**Table 4 ijms-22-13283-t004:** Primer sequences (Sigma Aldrich, Taufkirhen, Germany) of analyzed circRNAs, miRNAs and respective endogenous controls.

**RNAs**	**Stem Loop 5′-3′**	
miR-181b	CTCAACTGGTGTCGTGGAGTCGGCAATTCAGTTGAGACCCACCG	
miR-106a	CTCAACTGGTGTCGTGGAGTCGGCAATTCAGTTGAGCTACCTGC	
	**Forward 5′-3′**	**Reverse 5′-3′**
miR-181b	ACACTCCAGCTGGGAACATTCATTGCTGTCG	GTC GGC AAT TCA GTT GAG
miR-106a	ACACTCCAGCTGGGAAAAGTGCTTACAGTGC	
U6	GCTTCGGCAGCACATATACTAAAAT	CGCTTCACGAATTTGCGTGTCAT
hsa_circ_00001445	CTCCAAGATGGGCGAAAGTT	CAGATTCTGATCCACAAGCCTC
hsa_circ_00003028	CACTCTAGCCGAGAACTGTCC	TTGTCCTGTACTTCATGCGCTO
hsa_circ_00007915	GATCTTCGACAGCACAGAGCA	AGTTGGTGATGAGCCCTGC
hsa_circ_00008717	TCTGTCACGGCACTGGTTG	TCAGTTTCCGTAGATATCGCCC
β-actin	GTGGCCGAGGACTTTGATTG	CCTGTAACAACGCATCTCATATT

## Data Availability

The data presented in this study are available on request from the corresponding author at e-mail: maria.radanova@mu-varna.bg. The data are not publicly available due to their containing information that could compromise the privacy of research participants.

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
