# Peer review of "New Circulating Circular RNAs with Diagnostic and Prognostic Potential in Advanced Colorectal Cancer"

_ijms, 2021, doi:10.3390/ijms222413283_

Round 1

Reviewer 1 Report

Radanova and colleagues presented a research article aimed at assessing the diagnostic and prognostic potential of circRNA in colorectal cancer. For this purpose, the authors collected plasma samples from both CRC patients and healthy volunteers in order to establish the expression levels of four circRNA assessing their variation according to tumor disease and other clinical-pathological features. The authors have also predicted the prognostic values of such circRNAs and the expression levels of two targeted microRNAs. Overall, the manuscript is innovative and original, however, some improvements are needed:
1) In the introduction and discussion sections, the authors mentioned the potential role of lncRNAs as biomarkers. What about the small ncRNA. Several studies have demonstrated the high diagnostic and prognostic potential of microRNAs in different tumors, including colorectal cancer. Please add this further information. For this purpose, please see:
- PMID: 29779016
- PMID: 30801950
- PMID: 34208056
- PMID: 33066062
- PMID: 32911851
2) In some parts, English editing is recommended;
3) Please confirm the p-values obtained for the ROC curve. Both AUC and sensitivity/specificity values were not high, thus the p-values obtained from the authors seem too low;
4) Check the error “cicrRNAs” in the following title: “2.2 Comparison between the cicrRNAs plasma levels and the clinicopathological characteristics of the CRC patients”;
5) Please provide Figure 2 as it is not correctly displayed in the pdf file of the manuscript. Therefore, it is impossible to evaluate these data;
6) Why did the authors divide the samples according to hsa-circ_0001445 low and intermediate-high levels? They should divide samples into low and high (50:50 or 33:33 or 25:25);
7) In the Discussion or Introduction sections, please introduce other studies describing the potential diagnostic role of circulating circRNA in other tumors. For this purpose, see:
- PMID: 34198978
- PMID: 33816529
- PMID: 33335548
8) Consider shortening the Discussion section;
9) Please provide a table for the clinical-pathological data reported in Chapter 4.1.

Author Response

Reviewer 1

Radanova and colleagues presented a research article aimed at assessing the diagnostic and prognostic potential of circRNA in colorectal cancer. For this purpose, the authors collected plasma samples from both CRC patients and healthy volunteers in order to establish the expression levels of four circRNA assessing their variation according to tumor disease and other clinical-pathological features. The authors have also predicted the prognostic values of such circRNAs and the expression levels of two targeted microRNAs. Overall, the manuscript is innovative and original, however, some improvements are needed:

Comment1:

1) In the introduction and discussion sections, the authors mentioned the potential role of lncRNAs as biomarkers. What about the small ncRNA. Several studies have demonstrated the high diagnostic and prognostic potential of microRNAs in different tumors, including colorectal cancer. Please add this further information. For this purpose, please see:

- PMID: 29779016

- PMID: 30801950

- PMID: 34208056

- PMID: 33066062

- PMID: 32911851

Author’s Reply:

Thank you for this suggestion. We agree that miRNAs should not be forgotten, and have incorporated this information in the section "Introduction". The written paragraph, with respect to their role in cancer and potential as biomarkers, is small because otherwise, we are concerned that we would go out of focus on circRNAs in our manuscript. In this paragraph, we used all the references that you kindly provided us. All incorporated changes are presented using the ‘tracked changes’ function in the revised manuscript.

Comment 2:

2) In some parts, English editing is recommended

Author’s Reply:

Thank you for your comment. We improved the quality of the English language as much as we could and we believe that we now meet the high standards of the journal.

Comment 3:

3) Please confirm the p-values obtained for the ROC curve. Both AUC and sensitivity/specificity values were not high, thus the p-values obtained from the authors seem too low

Author’s Reply:

We confirm the p-values for the ROC curves. The values of AUC and sensitivity/specificity are presented in the cut-off on the largest Youden index for each circRNA, as we tried to explain in the manuscript in chapters 2.2 and 2.4. We used the Youden Index as an indicator of the maximum potential effectiveness of biomarkers. We think that it is a measure for a diagnostic test's ability to balance sensitivity and specificity. Youden Index gives us an opportunity to estimate the optimal cut-point for biomarkers. We could choose the cut-off that gave us more increased specificity or sensitivity of the biomarkers but we decided to present them in cut-off on the largest Youden indexes. Thus, sharing with our readers the specificity and sensitivity on the maximum Youden index we would allow them to assess if the test meets the empirical benchmarks for being administered for diagnostic purposes.

Comment 4:

4) Check the error “cicrRNAs” in the following title: “2.2 Comparison between the cicrRNAs plasma levels and the clinicopathological characteristics of the CRC patients”

Author’s Reply:

Thank you for you for noticing this technical mistake. It was revised and corrected accordingly.

Comment 5:

5) Please provide Figure 2 as it is not correctly displayed in the pdf file of the manuscript. Therefore, it is impossible to evaluate these data

Author’s Reply:

We apologize for this inconvenience. We did not notice that Figure 2 was not displayed because we submitted the manuscript as a word file template, as the submission system did not require to convert the manuscript into a *pdf file. Probably the other reason Figure 2 is not visible is that we did not paste it correctly in the template. We hope that this technical problem was avoided in the revised manuscript.

Comment 6:

6) Why did the authors divide the samples according to hsa-circ_0001445 low and intermediate-high levels? They should divide samples into low and high (50:50 or 33:33 or 25:25)

Author’s Reply:

We understand that data-dependent methods for dichotomizing continuous covariates, such as splits about some percentile (median, 25th, 75th), are arbitrary and may not be useful in assessing a variable’s true prognostic value (Altman et al., 1994). To overcome this issue we used an outcome-based method, which allowed us to separate low and high-risk patients with respect to overall survival (Mazumdar et al., 2000). We divided our patients in splits about percentiles (33th, 66th and 99th) in order to orient ourselves in the duration of their survival. This allowed us to assess the overall survival in all metastatic patients and to identify more precisely the group in which survival is most strongly influenced by the low expression level of the circRNA. We adopted this kind of assessment of the association between expression and survival in our work (Radanova et al., 2021) because it allows us to consider months of survival in the group of patients with an expression level of ncRNAs around the mean.

The representation of the expression as split about percentiles is not a precedent. The threshold for prognostic significance of ncRNAs is chosen manually also in other studies (Wach et al., 2015, Shukla et al., 2016, Siriwardhana et al., 2019).

References

  1. Altman, D. G., Lausen, B., Sauerbrei, W., & Schumacher, M. (1994). Dangers of using "optimal" cutpoints in the evaluation of prognostic factors. Journal of the National Cancer Institute, 86(11), 829–835. https://doi.org/10.1093/jnci/86.11.829
  2. Mazumdar, M., & Glassman, J. R. (2000). Categorizing a prognostic variable: review of methods, code for easy implementation and applications to decision-making about cancer treatments. Statistics in medicine, 19(1), 113–132. https://doi.org/10.1002/(sici)1097-0258(20000115)19:1<113::aid-sim245>3.0.co;2-o
  3. Radanova, M., Mihaylova, G., Mihaylova, Z., Ivanova, D., Tasinov, O., Nazifova-Tasinova, N., Pavlov, P., Mirchev, M., Conev, N., & Donev, I. (2021). Circulating miR-618 Has Prognostic Significance in Patients with Metastatic Colon Cancer. Current oncology (Toronto, Ont.), 28(2), 1204–1215. https://doi.org/10.3390/curroncol28020116
  4. Wach, S., Al-Janabi, O., Weigelt, K., Fischer, K., Greither, T., Marcou, M., Theil, G., Nolte, E., Holzhausen, H. J., Stöhr, R., Huppert, V., Hartmann, A., Fornara, P., Wullich, B., & Taubert, H. (2015). The combined serum levels of miR-375 and urokinase plasminogen activator receptor are suggested as diagnostic and prognostic biomarkers in prostate cancer. International journal of cancer, 137(6), 1406–1416. https://doi.org/10.1002/ijc.29505
  5. Shukla, S., Evans, J. R., Malik, R., Feng, F. Y., Dhanasekaran, S. M., Cao, X., Chen, G., Beer, D. G., Jiang, H., & Chinnaiyan, A. M. (2016). Development of a RNA-Seq Based Prognostic Signature in Lung Adenocarcinoma. Journal of the National Cancer Institute, 109(1), djw200. https://doi.org/10.1093/jnci/djw200
  6. Siriwardhana, C., Khadka, V. S., Chen, J. J., & Deng, Y. (2019). Development of a miRNA-seq based prognostic signature in lung adenocarcinoma. BMC cancer, 19(1), 34. https://doi.org/10.1186/s12885-018-5206-8

Comment 7:

7) In the Discussion or Introduction sections, please introduce other studies describing the potential diagnostic role of circulating circRNA in other tumors. For this purpose, see:

- PMID: 34198978
- PMID: 33816529
- PMID: 33335548

Author’s Reply:

Since this is the second comment on our description of the current state in ncRNAs field, we are sorry for our negligence of research progress representations. Considering your suggestion, we added information about the potential diagnostic role of circulating circRNAs in other tumors, using again the studies kindly provided by you. All incorporated changes are presented using the ‘track changes’ function in the section “Introduction” of the revised manuscript.

Comment 8:

8) Consider shortening the Discussion section

Author’s Reply:

We understand your concern about the length of the section "Discussion". Please allow us to explain why we would like to keep this section in its current form. In the “Discussion”, we tried to present our hypothesis concerning the origin of the investigated circRNAs. We believe this part of the manuscript is essential for understanding why did we observe its high expression in metastatic patients’ plasma. There are many unresolved questions regarding the difference in plasma and tissue expressions of ncRNAs in solid tumors and regarding the functions of circRNAs as sponges. Therefore, we tried to raise these questions in section "Discussion" to stimulate discussions in other subsequent studies.

Comment 9:

9) Please provide a table for the clinical-pathological data reported in Chapter 4.1.

Author’s Reply:

Thank you for this valuable suggestion. We made a Table A1 with clinicopathological data of patients and characteristics of all participants. Please, allow us to present it in Appendix because the new table includes the information which was described in detail in Section “Materials and Methods”, chapter 4.1 and it partially overlaps with some of the information presented in Table 2 in chapter 2.2.

We are very thankful for your observations and comments. We are hoping that we have understood your comments and our answers and the made corrections in the manuscript are acceptable.

Reviewer 2 Report

Overall I found this paper interesting and clinicaly valuable because of potential applicability of the results. I would like to address some comments to authors:

1. Authors enrolled only patients in stage III and IV therefore the title is a bit confusing, authors should address title to advanced CRC.

2. I am interested in that are there any differences in the tests accuracy between stage III and IV, separately for all studied molecules. Moreover it will be also interesting to combine circ and circ+miRNA into one diagnostic panel.

3. Authors found one circRNA involved in patients´ survival, please more carefully discuss why exactyl this molecule could be ralated to this event as well as the simple bioninformatics such as KEGG or GO are welcomed

4. Authors should more carefully explain why exactly these circRNAs were selected to the study protocol.

5. Figure no 2 is invisible, perhaps it is a fault of the file conversion, but should be checked before revision

Author Response

Reviewer 2

Overall I found this paper interesting and clinicaly valuable because of potential applicability of the results. I would like to address some comments to authors:

Comment 1:

  1. Authors enrolled only patients in stage III and IV therefore the title is a bit confusing, authors should address title to advanced CRC.

Author’s Reply:

Thank you for your recommendation. It was considered and changes were made in the title of the manuscript, accordingly:

"New Circulating Circular RNAs in Colorectal Cancer Patients with Diagnostic and Prognostic Potential"

changed to

"New Circulating Circular RNAs with Diagnostic and Prognostic Potential in Advanced Colorectal Cancer".

Comment 2:

  1. I am interested in that are there any differences in the tests accuracy between stage III and IV, separately for all studied molecules. Moreover it will be also interesting to combine circ and circ+miRNA into one diagnostic panel.

Author’s Reply:

Probably, because of the problem with the visualization of Figure 2 it was not clear that we found differences in AUCs between stage III and stage IV but only for hsa_circ_001445 and hsa_circ_0007915.

We compared the accuracy of hsa_circ_001445 and miR-181b and of hsa_circ_0007915 and miR-106a as diagnostic test. We found differences between AUCs of ROC curves only for first couple – hsa_circ_001445 and miR-181b (Difference=0.112, St. Error=0.044, Z-statistic=2,579, p=0.010). When we combined hsa_circ_001445 with miRNA-181b and performed ROC analysis we found that this panel could have better accuracy (AUC=0.929 95% CI: 0.884-0.974, p<0.001, Fig. 1, here). Unfortunately, the limited available number of patients (61) with data for miRNAs expression makes the results for such a panel unconvincing. This is a post-hoc analysis and it was not planned in our study it is only hypothesis-generating therefore let us not include it in the revisited manuscript. Also, it includes information only for metastatic patients. We could try to evaluate the circ+miRNA diagnostic panel/s in a larger cohort involving also patients at earlier stages of the disease in our next study.

Figure 1. Product of values for hsa_circ_001445 and miRNA-181b for the discrimination between healthy controls and metastatic CRC patients

Comment 3:

  1. Authors found one circRNA involved in patients´ survival, please more carefully discuss why exactyl this molecule could be ralated to this event as well as the simple bioninformatics such as KEGG or GO are welcomed

Author’s Reply:

Thank you for your comment. Unfortunately, at this stage of our study, we could only speculate why high expression of hsa_circ_0001445 was related to higher survival. We could comment on different other studies that performed functional analyses and investigated different target molecules. For example, Yu et al. (2018) have found that in hepatocellular carcinoma hsa_circ_0001445 sponges miR-17-3p and miR-181b-5p, thus promoting the expression of a well-known tumor suppressor of TIMP3 and a study by Tian & Liang (2018) have reported that hsa_circ_0001445 downregulates the cancerogenic miR-620 in cervical cancer.

There are no functional analyzes for hsa_circ_0001445 in CRC, which is a good opportunity to include such analyzes in our future research.

As to GO and KEGG analyses, we thank you for that proposal. Indeed, we are planning to use them to complement functional analyses in our next study. We did not present them in the current manuscript, because it was focused on the levels of investigated circRNAs and whether they have any significance for the clinics, do they have diagnostic and/or prognostic potential.

  1. Yu, J., Xu, Q. G., Wang, Z. G., Yang, Y., Zhang, L., Ma, J. Z., Sun, S. H., Yang, F., & Zhou, W. P. (2018). Circular RNA cSMARCA5 inhibits growth and metastasis in hepatocellular carcinoma. Journal of hepatology, 68(6), 1214–1227. https://doi.org/10.1016/j.jhep.2018.01.012
  2. Tian, J., & Liang, L. (2018). Involvement of circular RNA SMARCA5/microRNA-620 axis in the regulation of cervical cancer cell proliferation, invasion and migration. European review for medical and pharmacological sciences, 22(24), 8589–8598. https://doi.org/10.26355/eurrev_201812_16622

Comment 4:

  1. Authors should more carefully explain why exactly these circRNAs were selected to the study protocol.

Author’s Reply:

Thank you for your comment. Our work in the field of ncRNAs began with a study of the expression of circulating miRNAs in CRC patients. As a result, we detected several miRNAs with abnormal expression in patients’ plasma samples. In attempt to find out the reason of this, we examined our CRC patients for SNP in miRNA genes and whether there are any circRNAs, which act as sponge for miRNAs, involved in tumorigenesis. Therefore, we focus on the selected four circRNAs – three of them have never been studied in CRC before, and only one has been investigated in other studies with CRC patients. The results obtained for these circRNAs encourage us to summarize our findings in the current manuscript. We are planning to deepen the research on these and other circRNAs, in terms of their functionality, by including new analyzes. We believe that the analysis of circRNAs in the circulation has a future, since there is a small number of studies with liquid biopsy samples and even smaller studies with Caucasian cohorts.

Comment 5:

  1. Figure no 2 is invisible, perhaps it is a fault of the file conversion, but should be checked before revision

Author’s Reply:

We apologize for this inconvenience. We did not notice that Figure 2 was not displayed because we submitted the manuscript as a word file template. Probably the reason Figure 2 is not visible is that we did not paste it correctly in the template. We hope that this technical problem was avoided in the revised manuscript.

We are very thankful for your observations and comments. We are hoping that we have understood your comments and our answers and the corrections in the manuscript are acceptable.

Round 2

Reviewer 1 Report

The authors have significantly improved their manuscript answering well to all my comments. The revised version of the manuscript is now more complete and detailed. There are no further amendments, therefore, the manuscript can be accepted for publication after the editorial check.

Reviewer 2 Report

Authors responded for the all addressed comments. I do not have a new ones.